# Functionalization and Modification of Bitumen by Silica Nanoparticles

**Ainur Zhambolova [1], Anna Lisa Vocaturo [2], Yerbol Tileuberdi [1], Yerdos Ongarbayev [1,3], Paolino Caputo [4,*], Iolinda Aiello [2,5], Cesare Oliviero Rossi [4,*] and Nicolas Godbert [2]**

1 Faculty of Chemistry and Chemical Technology, Al-Farabi Kazakh National University, 71 al-Farabi Ave., 050040 Almaty, Kazakhstan; zhambolova.ainur@mail.ru (A.Z.); Erbol.tileuberdi@kaznu.kz (Y.T.); Erdos.ongarbaev@kaznu.kz (Y.O.)

2 MAT-InLAB, LASCAMM CR-INSTM, Unità INSTM della Calabria, Dipartimento di Chimica e Tecnologie Chimiche, Università della Calabria, Ponte Pietro Bucci Cubo 14C, 87036 Arcavacata di Rende (CS), Italy; anna.l.vocaturo@hotmail.com (A.L.V.); iolinda.aiello@unical.it (I.A.); nicolas.godbert@unical.it (N.G.)

3 Faculty of Energy, Oil and Gas Industry, Kazakh-British Technical University, 59 Tole bi Str., 050000 Almaty, Kazakhstan

4 Dipartimento di Chimica e Tecnologie Chimiche, Università della Calabria, 87036 Arcavacata di Rende (CS), Italy

5 CNR NANOTEC-Istituto di Nanotecnologia U.O.S. Cosenza, 87036 Arcavacata di Rende (CS), Italy

* Correspondence: paolino.caputo@unical.it (P.C.); cesare.oliviero@unical.it (C.O.R.)

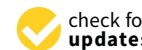

**Featured Application: Modification of the thermorheological properties of bitumen binder by incorporation of alkylated silica nanoparticles.**

**Abstract:** A study on the effect of silica nanoparticles (SNPs) dispersion in bitumen is herein reported. First, the size of the nanoparticles was finely tuned by controlling the experimental conditions during their synthesis, obtaining spherical SNPs with diameter ranging from 95 up to 900 nm. Subsequently, SNPs were embedded with peripheral amine groups by using APTES (3-aminopropyltriethoxysilane) as functionalized agent ($NH_2$@SNP), and ultimately long alkyl chains were grafted by reacting the free amine with an alkylated aldehyde ($C_{14}N$@SNP). All SNPs (ca. 1 wt%.) were dispersed in bitumen to probe their effect on the rheological properties of bitumen. No significant change in the thermorheological properties of bitumen was observed upon varying the size of the SNPs. Slight improvement was observed when using $NH_2$@SNPs, while the best results were obtained by using $C_{14}N$@SNPs, showing the crucial role that hydrophobic substituents play in bitumen binders which leads to significant improvements.

**Keywords:** bitumen; silica nanoparticles; thermorheology; inorganic-organic composite

---

## 1. Introduction

One of the methods for improving the characteristics of bitumen binders is the modification brought about by the use of nanomaterials [1]. The addition of nanomaterials indeed changes the rheological properties of bitumen and can also lead to changes in intermolecular forces in the structure of bitumen binders. This is due to the fact that as the dimensions of the materials reach the nanoscale, interactions at the phase boundaries become much more intense [2].

In this context, silica nanoparticles (SNPs) are among the nanomaterials that are proposed to be used as additives in asphalt mixtures [3]. It was recently shown that the addition of nanosilica leads to an increase in rigidity, tensile strength, elastic modulus, fatigue life, and resistance to permanent deformation and damage from moisture in asphalt concrete mixtures [4]. In particular, evaluation of the characteristics of bitumen incorporating SNPs showed a gain in high fatigue resistance, also

improving oxidative aging and rutting properties [5–7]. Furthermore, a study of the effect of SNPs on the thermal properties of bitumen binders showed that nanosilica could improve the temperature sensitivity of asphalt mixtures [8].

Bitumen is an organic binder formed by high molecular weight hydrocarbons and their derivatives. The contrast between an organic based matrix and an inorganic additive might result in a poor compatibility resulting in a scarce dispersibility of the nanoparticles within bitumen [9]. To improve the dispersion of nanosilica in organic matrices and favor interaction with the medium, functionalization of the surface of nanosilica can be used. In this way, the number of silanol structures are reduced changing the structure of surface functional groups thus mitigating the mismatch between nanosilica particles and the bitumen matrix [10]. Thus, surface properties such as the physicochemical adsorption of nanosilica and the surface free energy of the nanoparticles can be adequately altered, and the agglomeration phenomenon between the particles inside the matrix decreases.

A recent study reported that $NH_2$@SNP: silica nanoparticles functionalized with APTES (3-aminopropyltriethoxysilane), can effectively improve bitumen resistance to oxidation and mechanical stresses, while reducing the agglomeration phenomena observed with pristine SNPs. Up to 4 wt% of modified SNPs were dispersed in bitumen without observing any agglomeration process [11].

The present study intends therefore to use the free amine moieties of APTES-modified SNPs as functional groups for further alkylation derivatization and check the eventual improvements of such modifications onto the thermorheological properties of bitumen. To this end, alkylated SNPs ($C_{14}$N@SNP) were synthesized with the aim of increasing the hydrophobicity of the nanoparticles, thus improving their dispersion in the bitumen binder, as well as enhancing its thermorheological properties.

However, before testing our newly functionalized SNPs, a curiosity arose: does the size of the SNPs have any influence on the rheological properties of bitumen? To answer this question, a series of SNPs were first prepared by varying the experimental conditions in order to finely control the diameter of the SNPs produced. Indeed, in the sol-gel method the final size of SNPs is greatly affected by different parameters, such as the chemical nature of the alcoholic solvent, the alcohol/water ratio, the silica precursor, the catalyst, the surfactant concentration and the reaction temperature [12,13]. In this study, SNPs of different size were obtained by varying two parameters in turn: the ethanol/water ratio and the reaction temperature. Finally, the comparison between surface-functionalized and pristine SNPs on the rheological properties of bitumen was conducted.

## 2. Experimental

### 2.1. Materials and Reagents

Cetyltrimethylammonium bromide (CTAB), 3-aminopropyltriethoxysilane (APTES), tetraethylorthosilicate (TEOS) and tetradecyl aldehyde were purchased from Sigma Aldrich (Milano, Italy). The bitumen used in this work was kindly supplied by Loprete Costruzioni Stradali (Terranova Sappo Minulio (RC), Italy). It was produced in Italy but the crude oil was gotten from Saudi Arabia (Table 1). Its penetration grade (PN) was 50/70. The main empirical bitumen characteristics are reported in Table 1.

**Table 1.** Properties of pristine bitumen.

| Physical Quantity | Procedure | Value |
|---|---|---|
| PN (0.1 mm) Penetration depth ± 1 | ASTM D 946 | 68 |
| R&B (°C) Softening point ± 1 | ASTM D 36 | 46.8 |
| TR (°C) Transition temperature ± 0.1 | DSR [14] | 60.5 |

### 2.2. Synthesis

**SNPs**: 0.24 g of CTAB was dissolved in a given amount of absolute ethanol and distilled water, under stirring with a magnetic stirrer. When CTAB was completely dissolved, 3 mL of ammonia solution (25 wt% in water) and 5.4 mL of TEOS were added in sequence. Few minutes after the addition of TEOS,

the clear solution turned white indicative of the formation of precipitated SNPs. The resulting suspension was vigorously stirred for 2 h. The product was separated by filtration and washed with water (2 × 50 mL) and ethanol (2 × 50 mL). The crude powder was dried at 65–80 °C for about 2 d, and finally calcined in air at 650 °C for 3 h. Reaction temperature and the relative ratio of ethanol/water were varied to produce silica nanoparticles of different sizes (SNPs 1–4). See also Table 2 for complementary information in particular the ethanol/water ratios used and the temperature at which the reaction was conducted.

$NH_2$@**SNP-3**: 1.4 g of **SNP-3** were dispersed in 1 L of ethanol, by sonication. Then, 12 mL of APTES was added and stirred at room temperature overnight. The product was centrifuged and washed five times with ethanol. The obtained washed precipitate was dried at 60 °C for 20 h in air and then under vacuum.

$C_{14}N$@**SNP-3**: 1.7 g of $NH_2$@**SNP-3** was dispersed in 1 L of ethanol, by sonication. Then, 0.2 g of tetradecyl aldehyde was added under stirring and the mixture was refluxed for 1 h, then cooled and filtered. The product was abundantly washed with THF (6 × 50 mL), dried under vacuum 1 h, and placed in the oven at 65–70 °C for about 3 d.

### 2.3. Preparation of Modified Bitumen with Silica Nanoparticles (SNPs)

Each additive was mixed with hot bitumen (140–160 °C) in the 1 wt% ratio. The bitumen was modified by using a mechanical stirrer (IKARW20, Germany). Firstly, 100 g of bitumen was heated up to 140–160 °C until it fully flowed, then a given amount of additive was added to the melted bitumen under a high-speed shear mixer of 600–700 rpm. Furthermore, the mixture was stirred again at 140–160 °C for 30 min. After mixing, the resulting bitumen mixture was poured into a small sealed can and then stored in a dark chamber thermostated at 25 °C to retain the obtained morphology as previously performed in other studies [15].

### 2.4. SNPs Characterization

Sonication used to disperse nanoparticles during the synthesis step was achieved by using a UP200S ultrasonic Hielscher processor equipped with a sonotrode of 40 mm. Thermogravimetric analysis was performed on a PerkinElmer PYRIS 6 TGA thermogravimetric analyzer. Fourier transform infrared spectroscopy (FT-IR) was recorded on a PerkinElmer Spectrum 100 spectrometer. Rheological measurements were carried out using an SR5000 rheometer (Rheometrics, Piscataway, NJ, USA) controlled by shear stress and equipped with plate geometry (2 mm gap, 25 mm diameter). The temperature was controlled by a Peltier system (±0.1 °C). The rheological behavior at different temperatures was investigated using a time cure test at 1 Hz with a ramp rate of 1 °C/min within the linear viscoelastic region. [16] Scanning electron microscopy (SEM) was undertaken using an environmental SEM (ESEM; FEI Quanta 400; Hillsboro, OR, USA).

## 3. Results and Discussion

### 3.1. Influence of the Diameter Size of SNPs

All **SNPs** were prepared by following a modified Stöber method. In particular, cetyltrimethylammonium bromide (CTAB) was utilized as template agent to obtain spherical-shaped **SNPs** (Scheme 1).

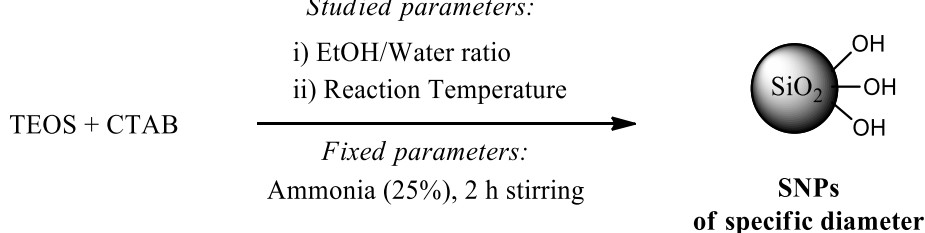

**Scheme 1.** Synthesis of the studied silica nanoparticles (SNPs).

By varying the experimental conditions, SNPs of different sizes were prepared (**SNP1-4**). The influence of two parameters was studied, relative content of ethanol/water of the reaction medium and temperature at which the reaction took place. Experimental conditions and obtained results are presented in Table 2. The average size was determined by SEM microscopy.

**Table 2.** Preparation of SNPs of different sizes.

|  | Variation in Ethanol/Water Volume Ratio | | | | Effect of Temperature | |
|---|---|---|---|---|---|---|
| SNP Sample | SNP-1 | SNP-2 | SNP-3 | SNP-4 | SNP-3a | SNP-3b |
| EtOH (mL) | 39 | 52 | 78 | 156 | 78 | 78 |
| $H_2O$ (mL) | 204 | 191 | 165 | 88 | 165 | 165 |
| T ($^\circ$C) | 25 | 25 | 25 | 25 | 40 | 50 |
| Average Size (nm) | 95 ± 20 | 150 ± 25 | 300 ± 55 | 700 ± 45 | 500 ± 55 | 900 ± 65 |

Note that the average sizes and standard deviations of the SNPs were determined through scanning electron microscopy (SEM) by observing, for each sample, an area containing ca. 500 nanoparticles.

With the temperature increase (from 25 to 50 °C), the size of the SNPs increases (from 300 to 900 nm, samples **SNP-3**, **SNP-3a** and **SNP3b**), similarly the diameter of the nanoparticles also increases by increasing the amount of ethanol with respect to water (Samples **SNP-1–4**). Thus playing with these two parameters allowed us to obtain 6 samples of nanoparticles with diameter size ranging from ca. 100 nm to 1 μm (Table 1). SEM images of the synthesized **SNPs** are reported in Figure 1. The experimental conditions adopted here allowed us to obtain spherical shaped nanoparticles, however, as observed in Figure 1a,b, for the smallest size nanoparticles, their morphology tended to be more similar to flakes than true spheres.

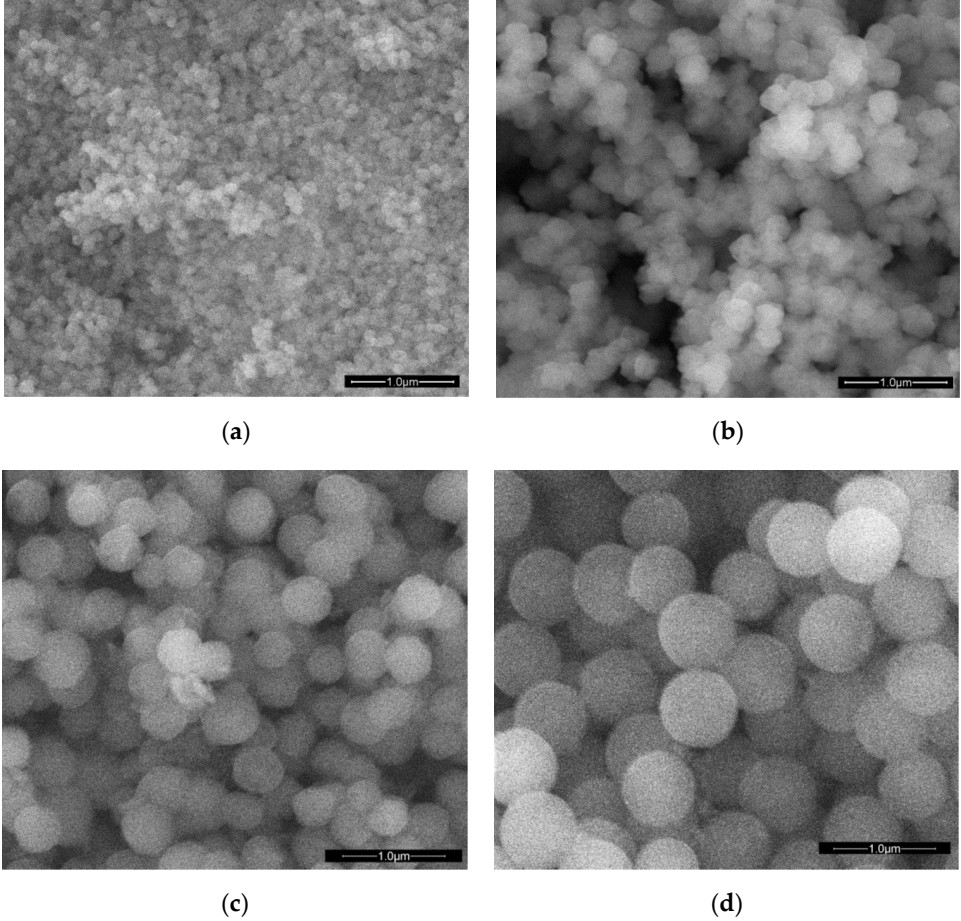

(a)

(b)

(c)

(d)

**Figure 1.** *Cont.*

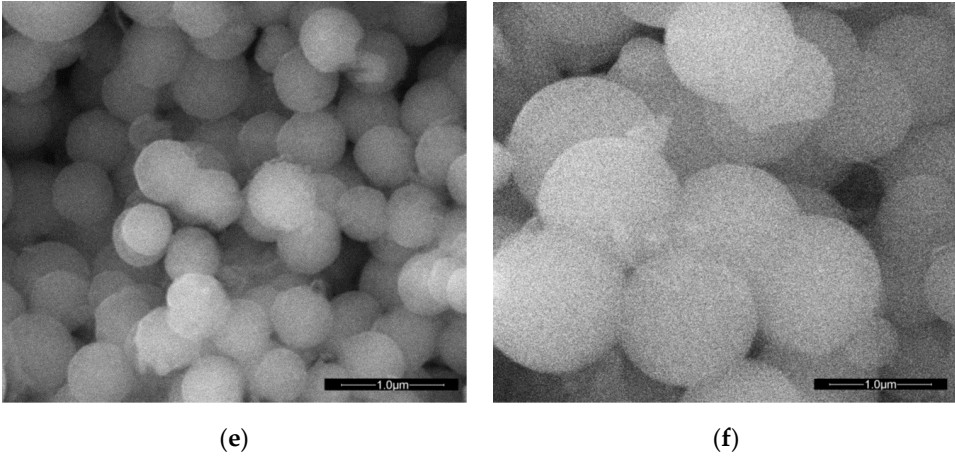

(**e**)          (**f**)

**Figure 1.** SEM images of synthesized SNPs. (**a**) SNP-1, (**b**) SNP-2, (**c**) SNP-3, (**d**) SNP-4, (**e**) SNP-3a and (**f**) SNP-3b. Note that all images were taken at identical magnification to allow immediate comparison (scale bar: 1 μm).

The influence of the size of the nanoparticles on the rheological properties of bitumen was probed by incorporating all SNPs in bitumen (1% weight ratio). This low ratio was chosen according to the literature [16] keeping in mind that the main goal in asphalt binder uses is to reduce at the maximum the quantity of additive for economic purposes but also to minimize its eventual aggregation that could result in sedimentation phenomenon over a long-term period. Obtained results in terms of the time cure test (tan δ versus temperature) are reported in Figure 2. The performed analysis showed that the size of the nanoparticles with diameter ranging from 100 nm to 1 μm, does not have any significant influence on the rheological properties of the bitumen. All additives, independent of their size act as an inert filler such as $CaCO_3$, that we used and tested in the same experimental conditions for direct comparison. Consequently, smaller size nanoparticles are preferred in order to minimize over time eventual sedimentation phenomena, in fact, effects of sedimentation in the bitumen related to the particles size are well known in literature [17].

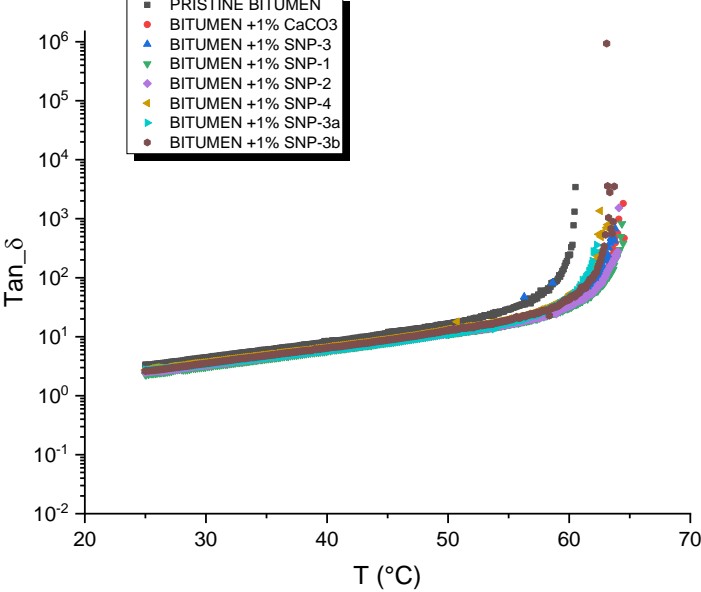

**Figure 2.** Semi-log plot of temperature ramp tests for the pristine and modified bitumens formulated with 1 wt% of nanomaterials added as solid powder dispersion. The effect of an inert filler $CaCO_3$ (1 wt%) was also determined. Loss tangent tan δ is plotted, the asymptotic value identifies the sol-transition temperature.

### 3.2. Functionalization of SiO₂ Nanoparticles.

Having checked that the **SNPs'** dimension has no significant influence on the rheological properties of bitumen, we functionalized the surface of the **SNPs** differently to probe the effect of peripheral functional groups. Chemical modifications of the SNPs were performed as described in Scheme 2.

**Scheme 2.** Reaction scheme of the functionalization of SNPs.

First, **SNP-3** was reacted with 3-aminopropyltriethoxysilane (APTES) after careful dispersion in methanol through intense sonication process. After 24 h of reaction time, the resulting solution was centrifuged and the resulting residue was thoroughly washed with methanol and dried to yield **NH₂@SNP-3** nanoparticles embedded with free NH₂ pending groups. In a second step, the NH₂@SNPs obtained were re-dispersed in ethanol and after a strong sonication process, reacted with tetradecyl aldehyde in order to obtain the **C₁₄N@SNP-3** nanoparticles embedded with $C_{14}$ alkyl length chains. SEM images of **SNP-3** and the corresponding functionalized **NH₂@SNP-3** and **C₁₄N@SNP-3** samples are reported in ESI. All samples show rather identical features with nanoparticles similar in size and size distribution highlighting that no significant increase in diameter was observed with chemical functionalization. The success of the reactions was established by coupling the FT-IR spectra with the thermogravimetric curves obtained for all compounds. All FT-IR spectra are reported in Figure 3. The FT-IR spectrum of silica particles **SNP-3** showed an intense peak at 1100 cm⁻¹ and a wide weak peak in the range between 3200 and 3500 cm⁻¹, which is associated with vibrations of the Si-O-Si network and the stretching vibration of the peripheral OH groups (-Si-OH) bonds, respectively. The absorption band at 1637 cm⁻¹ is characteristic of the OH bending vibration. The FT-IR spectrum of the amine-modified silica sample **NH₂@SNP-3** showed in contrast the presence of a small broad peak at 2931 cm⁻¹, associated with the stretching vibrations of the saturated C-H bonds of the alkyl chains of the grafted APTES moieties. Although a wide band is observed at 3100–4000 cm⁻¹, the expected double amine absorption band that should be observed in this region cannot be distinguished clearly, probably due to the presence of Si-OH groups suggesting an incomplete functionalization of the **SNPs** with APTES. This is further confirmed by the absorption band at 1637 cm⁻¹ being still present although of less relative intensity when compared to the FT-IR spectrum of SNP-3 sample. Finally, the FT-IR spectrum of the **C₁₄N@SNP-3** clearly shows an increment in intensity and a sharpening of the bands at 2931 cm⁻¹ and 2929 cm⁻¹ relative to stretching vibration modes of the saturated C-H bonds of the long alkyl chains, accompanied by an increment in intensity of the band at 1000–1100 cm⁻¹ relative to the carbon skeleton of the alkyl chains, which overlap the vibration band associated with the Si–O–Si network. Although the stretching vibration of the C=N imine bond that should be observed at ca. 1650 cm⁻¹ is not clearly visible, probably overlapped with the 1637 cm⁻¹ bending vibration of the peripheral Si–OH residual groups, these modifications of the FT-IR features reveals the successful conversion of the NH₂ peripheral groups.

To confirm the success of the functionalizations performed and estimate the quantity of functional groups grafted onto the surface of the SNPs, thermogravimetric analyses were performed on all samples. TGA curves are reported in Figure 4.

The TGA trace of **SNP-3** is characterised by an initial loss in weight (ca. 3%) at temperature below 100 °C owing to the dehydration of the nanoparticles, before reaching a plateau correlated to the thermal stability of the silica nanoparticles up to 900 °C as expected.

Instead, the **NH₂@SNP-3** thermogravimetric curve is characterized by a much more pronounced initial weight loss owe to dehydration of ca. 8%. This is to be correlated to the more hygroscopic

character of the NH$_2$ superficial moieties with respect to the hydroxyl groups of the unfunctionalized **SNP-3** initial compound. With an increase in temperature, a second weight loss of ca. 6% starting at 300 °C is observed, which is associated with the thermal decomposition of the organic component, i.e., the aminopropyl chains. The thermogravimetric curve of **C$_{14}$N@SNP-3** in contrast displays a more hydrophobic character shown by the small amount of water molecules evaporated during dehydration (ca. 2% of weight loss) and a much pronounced weight loss starting at 300 °C. The latter corresponds to ca. 15.4% of loss owing to the presence of a larger amount of organic component, in agreement with the increase of the organic part and therefore evidence of the success of the alkylation reaction.

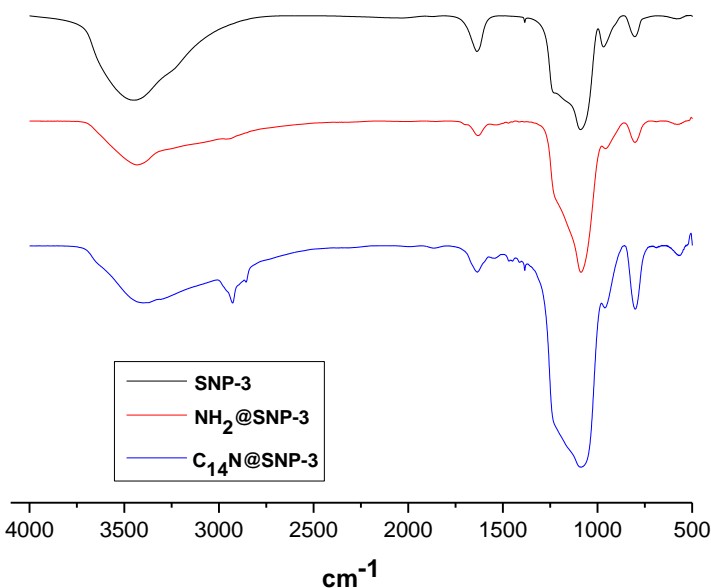

**Figure 3.** Fourier transform infrared (FT-IR) spectra of **SNP-3**, **NH$_2$@SNP-3** and **C$_{14}$N@SNP-3**. Note that samples were dried in oven at 60 °C overnight prior to analysis.

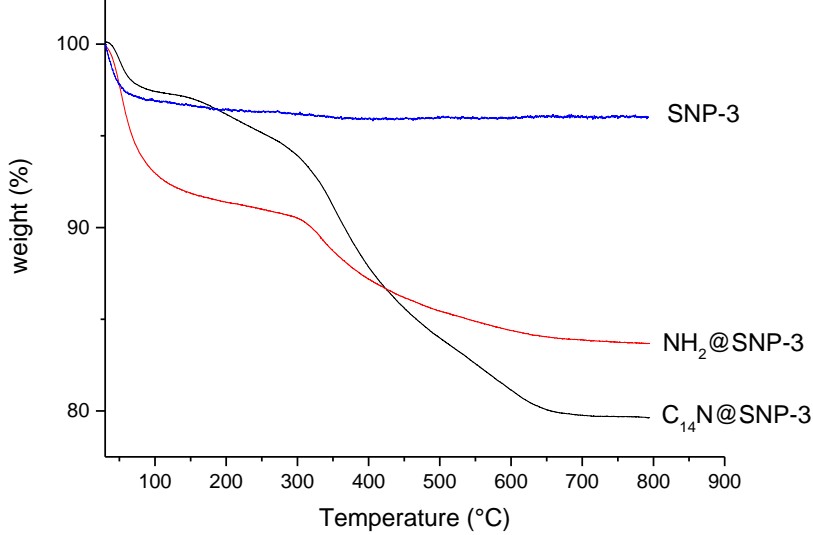

**Figure 4.** Thermogravimetric analysis (TGA) traces of SNP-3, NH$_2$@SNP-3 and C$_{14}$N@SNP-3.

To study the effect of functionalized silica particles on the properties of bitumen, the rheological properties of bitumen and functionalized **SNPs** (**FSNPs**) modified bitumen were determined. Functionalized silica particles **NH$_2$@SNP-3** and **C$_{14}$N@SNP-3** (ca. 1% in weight) were used to modify the bitumen according to the procedure previously described in Section 2.3.

Figure 5 shows the temperature dependencies of the elastic modulus and loss tangent of the pristine bitumen and **FSNP**s modified bitumen in the temperature range of 20–90 °C. As can be seen from the results, for pristine bitumen, a decrease in the elastic modulus and an increase in the loss tangent are observed up to 60 °C, while for a bitumen sample modified with **NH$_2$@SNP-3** particles, the dependency curves of viscoelastic properties decrease to 66–67 °C. At 60 °C, the elastic modulus is reduced to values of about $10^3$ Pa that differ from the values 10 Pa obtained for pristine bitumen (two orders of magnitude). These data confirm the change in the viscoelastic properties of bitumen as a result of modification.

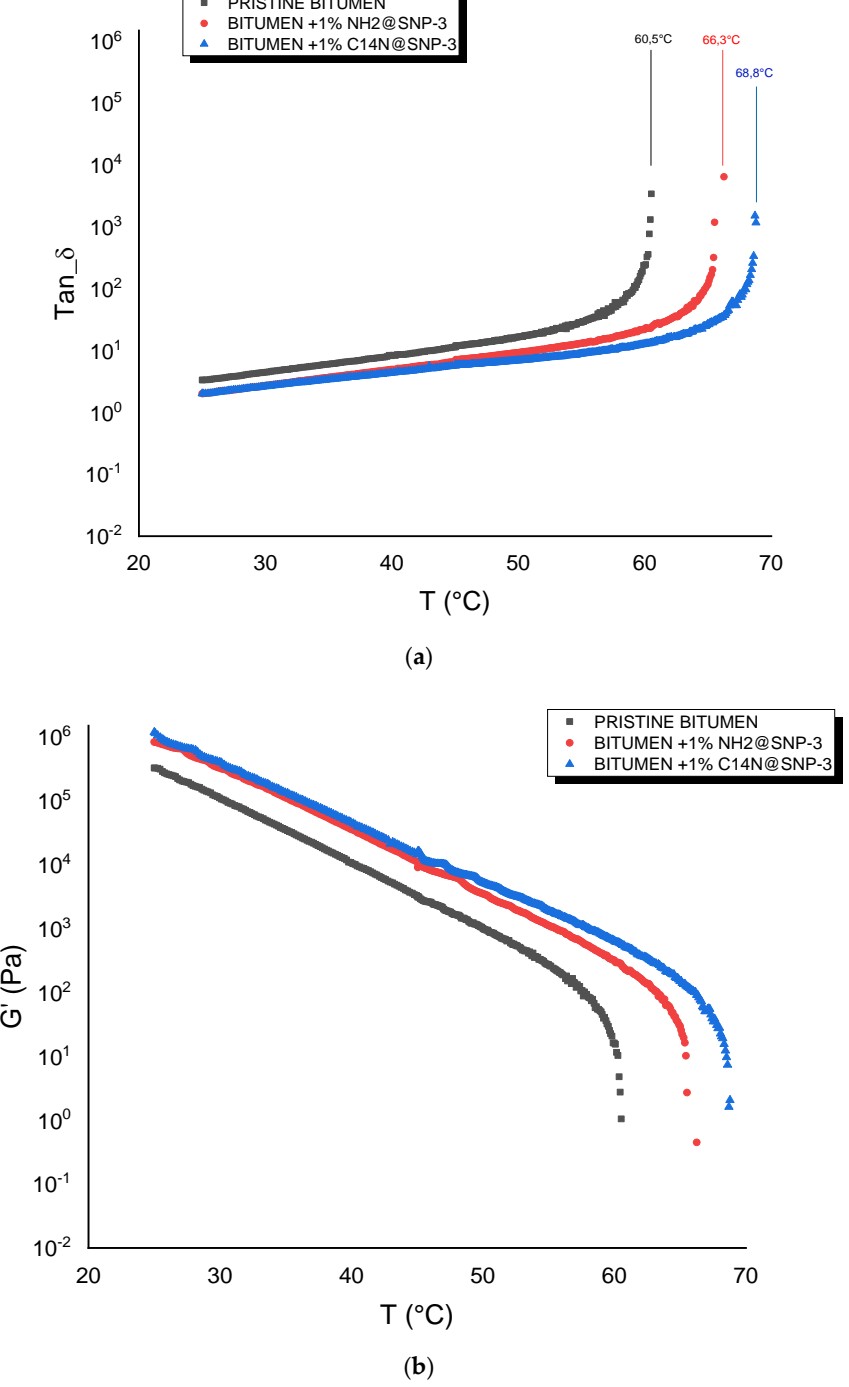

**Figure 5.** Temperature dependencies of the loss tangent (tan δ) (**a**) and elastic modulus (G′) (**b**) of the initial bitumen and after modification with 1% of **NH$_2$@SNP-3** and **C$_{14}$N@SNP-3**. Arrows indicate phase transition temperatures.

The temperature dependencies of the elastic moduli and the loss tangent of bitumen modified by particles of $C_{14}N@SNP-3$ in the amount of 1 wt% differ significantly [18,19]. It is worth noting that when 1% of $C_{14}N@SNP-3$ particles are added, the elastic modulus curves decrease to a temperature of 70 °C, and at this temperature the loss modulus reaches a vertical asymptote. These values are higher than those observed for the $NH_2@SNP-3$ modified bitumen.

Thus, when modifying bitumen, a change in rheological characteristics is observed depending on the type of modifier. A comparison of the temperature dependencies of the dynamic parameters before and after modification showed that the addition of functionalized silica particles leads to a shift of the curves to higher temperatures, which confirms a qualitative change in the physico-mechanical characteristics of bitumen. At high temperatures, the elastic modulus G′ has a non-linear behavior approaching the transition of a viscoelastic fluid into a fluid (60 and 70 °C for pure and modified bitumen, respectively). The transition from the viscoelastic to the liquid mode ends when the modulus G′ is no longer detected and, therefore, the loss tangent tan δ diverges at approximately the same temperature values [20]. The loss tangent tan δ increases with increasing temperature, which indicates a decrease in the consistency of the material and the predominant liquid-like behavior increases with increasing temperature.

It is noteworthy that a hardening effect marked by a shift of the sol-transition temperature higher than 5 °C is rather significant for bitumen. Similar rheological behaviors were receded for well-known additives such as polyphosphoric acid (PPA) [21,22] but environmental contamination issues have to be taken into account when using such additives. To this regard, FSNPs are far more eco-friendly additives. In this context, the recently published work of Karnati et al. [23] also shows the performances of diversely functionalized **SNPs** with long alkyl chains embedded with silane end groups to improve anti-aging and low temperature properties. Although these **FSNPs** require multi-step synthesis and were used in 4% weight ratio, they also show the importance of functionalized nanoparticles to promote their application in road paving with enhanced performance and sustainability.

Considering the micellar method usually adopted for bitumen description, organic molecules have been proven to stabilize clusters of a wide variety of inorganic nanoparticles [24,25]. Therefore, in this work the stabilization of nanoparticles is carried out using the principle of altering the nanoparticle Surface free energy [26,27], driven to the extreme with chemical functionalization of nanoparticles. However, in the complex framework of the micellar model, it can be argued that another nanoparticle stabilization mechanism is concurrently present, i.e., the competitive interactions between the organic moiety of the capping agent and the aphiphilic resins pre-existent in the bitumen. In fact, it has been recently highlighted that specific interactions between surfactants and organic molecules can give peculiar self-assembly processes [28] with consequences in dynamical behaviour and transport properties [29,30]. Therefore, the final result must be seen as the delicate summing up of different contributions coming from nanoparticles, from the capping agent's nature, and their interactions with the complex amphiphilic molecules of the bitumen, all embedding the apolar matrix.

## 4. Conclusions

Bared and functionalized silica nanoparticles were successfully dispersed in bitumen and their effects on the mechanical performance of the bitumen were investigated in depth. The size of the bared nanoparticles was controlled by modifying the experimental conditions during their synthesis (temperature and ethanol/water ratio), obtaining spherical nanoparticles of diameters ranging from ca. 100 nm to almost 1 μm. When mixed with bitumen (1 wt% ratio), the rheological properties of bitumen are almost unaltered, independent of the size of the nanoparticles. Instead, chemical modification of the surface of the nanoparticles by grafting long alkyl chains proved to be an efficient methodology to improve the mechanical performance of the bitumen. $NH_2$ free end groups were introduced first by reacting the nanoparticles with APTES, and in a second step, long alkyl chains were introduced by reaction of the free amine groups with tetradecyl aldehyde. When still using such a low amount of nanoparticles (1% weight) in bitumen, the introduction of $NH_2$ groups improves the mechanical



performance of the asphalt, but better results were observed for the alkylated nanoparticles. These results confirm the effectiveness of functionalized silica nanoparticles as modifiers of bitumen. The shift of the transition from viscoelastic to liquid phase is significant. The comparison of the viscoelastic trends with the ineffective $CaCO_3$ filler indicates clearly that 1 wt% can be considered a sufficient dosage for bitumen modification. The present study thus paves the way for the use of functionalised silica nanoparticles as a new additive for road pavements.

**Author Contributions:** Conceptualization, N.G.; Writing-Original draft preparation I.A. and N.G.; Writing-review and Editing Y.T. and Y.O.; Formal analysis Y.O.; Investigation A.Z. and A.L.V.; Supervision P.C. and C.O.R. All authors have read and agreed to the published version of the manuscript.

**Funding:** This work was partially supported by Loprete Costruzioni Stradali (Terranova Sappo Minulio (RC), Italy).

**Acknowledgments:** The authors greatly acknowledge Giovanni Desiderio (IMIP-CNR laboratory) for the acquisition of SEM images.

**Conflicts of Interest:** The authors declare no conflict of interest.

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
