# Peer review of "Functionalization and Modification of Bitumen by Silica Nanoparticles"

_applsci, doi:10.3390/app10176065_

Round 1

Reviewer 1 Report

Comments to the authors:

This paper reported the modification of bitumen by functionalized silica nanoparticles. Synthesized silica nanoparticles were characterized using SEM, FTIR and thermogravimetric techniques. The properties of modified bitumen were investigated. The manuscript can be considered for publication after major corrections that should be revised by the authors.

Specific comments:

  • The English language and style must be checked. For instance, It is very difficult to read Introduction part. Sometimes the meaning of the text is not clear (For instance in 52-54 lines: “In addition, bitumen is an organic binder formed by high molecular weight hydrocarbons and their derivatives, which can characterize nanosilica in a bitumen binder with poor dispersibility and compatibility.”).
  • Lines 51-52: “However, nanosilica, as a variety of inorganic nonmetallic nanomaterials, is very prone to agglomeration.” I do not agree with this statement. The agglomeration of all types of nanoparticles depends on various factors. It is possible to produce very stable Si nanoparticles, which are commercial available.
  • Why authors used methanol during modification of SNP-3 nanoparticles? Less harmful organic solvent can be used for this purpose. For instance, methanol can be replaced by ethanol, which was used for further modification.
  • Average size of nanoparticles in Table 2 should be presented with standard deviation.
  • It is not clear enough why SNP-3 nanoparticles were chosen for further modification. Also, the abbreviation C14N@SNP was used in introduction part, while throughout the manuscript it was called C14N@SNP-3. Unification of abbreviations are needed.
  • The conclusions should be improved. Obtained results should be structured. Please provide short conclusion with accurate and clear description of obtained results.

Author Response

Thank you for sending the reports of Reviewers 1 and 2. Please find our point-by-point replies to their observations, which are ordered in the way they posed the comments. Changes to the original manuscript are highlighted in yellow in the revised form. The authors are grateful to the Reviewers for their useful suggestions that helped to improve the present manuscript.

REFEREE 1

  • The English language and style must be checked. For instance, It is very difficult to read Introduction part. Sometimes the meaning of the text is not clear (For instance in 52-54 lines: “In addition, bitumen is an organic binder formed by high molecular weight hydrocarbons and their derivatives, which can characterize nanosilica in a bitumen binder with poor dispersibility and compatibility.”).

Answer: English language has been checked and corrected. Some sentences have been shorten, All modifications have been highlighted within the main text.

  • Lines 51-52: “However, nanosilica, as a variety of inorganic nonmetallic nanomaterials, is very prone to agglomeration.” I do not agree with this statement. The agglomeration of all types of nanoparticles depends on various factors. It is possible to produce very stable Si nanoparticles, which are commercial available.

Answer: Thanks for the referee for its observation. The sentence was badly written leading to this error in statement. Indeed, the authors wanted to point out that in a matrix such has bitumen the unfavorable interactions between all-inorganic nanoparticles and the matrix might more easily prompt to agglomerations of the nanoparticles. The sentence has been modified within the main text.   

  • Why authors used methanol during modification of SNP-3 nanoparticles? Less harmful organic solvent can be used for this purpose. For instance, methanol can be replaced by ethanol, which was used for further modification.

Answer: Thanks for the referee for pointing out this error and sorry for the mistake, indeed ethanol has been used as for the following step. Correction has been made.

  • Average size of nanoparticles in Table 2 should be presented with standard deviation.

Answer: Standard deviations have been included in the reported data table. Deviations have been determined through SEM images taken at different magnifications according to the size of the nanoparticles, in order to be able to observe around 500 nanoparticles for each sample.

  • It is not clear enough why SNP-3 nanoparticles were chosen for further modification. Also, the abbreviation C14N@SNP was used in introduction part, while throughout the manuscript it was called C14N@SNP-3. Unification of abbreviations are needed.

Answer: Having demonstrated that the size of the nanoparticles does not have any influence onto the rheological properties of bitumen, any size could have been employed for the following part of our study. We chose the SNP-3, being for us the “standard preparation” from which parameters have been modified to finely tune the dimension of the nanoparticles. Unification of the abbreviations have been performed.  

  • The conclusions should be improved. Obtained results should be structured. Please provide short conclusion with accurate and clear description of obtained results.

Answer: Conclusions have been accordingly revised

Reviewer 2 Report

Silica nanoparticles and microparticles were prepared by authors and utilized in composite with bitumen. According to the reaction conditions it was possible to tune the size of silica and finally these were modified by amine groups and by Schiff bases. The particles were characterized by SEM, thermogravimetry and IR spectroscopy. Rheological properties of composites are reported. Incorporation of modified silica particles have influence on asphalt properties.

The instruments suppliers should be better described. In description of IR spectra one should expect vibrations of amine and Schiff base are given. It seems that there is weight increase above cca 500 degrees at the TGA curve of  SNP-3. Can this be explained? In Fig. 5, a and b are moved. In References, the journals have shortcuts and sometimes all the titles are mentioned.   

Author Response

Thank you for sending the reports of Reviewers 1 and 2. Please find our point-by-point replies to their observations, which are ordered in the way they posed the comments. Changes to the original manuscript are highlighted in yellow in the revised form. The authors are grateful to the Reviewers for their useful suggestions that helped to improve the present manuscript.

REFEREE 2

Silica nanoparticles and microparticles were prepared by authors and utilized in composite with bitumen. According to the reaction conditions it was possible to tune the size of silica and finally these were modified by amine groups and by Schiff bases. The particles were characterized by SEM, thermogravimetry and IR spectroscopy. Rheological properties of composites are reported. Incorporation of modified silica particles have influence on asphalt properties.

The instruments suppliers should be better described. In description of IR spectra one should expect vibrations of amine and Schiff base are given. It seems that there is weight increase above cca 500 degrees at the TGA curve of  SNP-3. Can this be explained? In Fig. 5, a and b are moved. In References, the journals have shortcuts and sometimes all the titles are mentioned.   

Answer: Instrument suppliers have been improved throughout the main text. The slight increase that was observed for the SNP-3 TGA curve above 500°C was only due to a deviation of the baseline for this specific measurement. we therefore performed a new measurement on SNP-3 sample taking care of recording a new baseline prior to the sample measurement. The new curve thus obtained does not show any weight increase. Thanks to the referee for pointing this fact that could have mislead the reader to unfounded conclusions. Captions of Fig.5 have been corrected. Mistakes in references have been checked and properly addressed.

Round 2

Reviewer 1 Report

This paper can be published in the present form.